# On the alignment of velocity and magnetic fields within magnetosheath jets

Ferdinand Plaschke[1], Maria Jernej[1], Heli Hietala[2,3,4], and Laura Vuorinen[3]

[1]Space Research Institute, Austrian Academy of Sciences, Graz, Austria.
[2]The Blackett Laboratory, Imperial College, London, UK.
[3]Department of Physics and Astronomy, University of Turku, Turku, Finland.
[4]Department of Earth, Planetary, and Space Sciences, University of California Los Angeles, CA, USA.

**Correspondence:** Ferdinand Plaschke (ferdinand.plaschke@oeaw.ac.at)

**Abstract.** Jets in the subsolar magnetosheath are localized enhancements in dynamic pressure that are able to propagate all the way from the bow shock to the magnetopause. Due to their excess velocity with respect to their environment, they push slower ambient plasma out of their way, creating a vortical plasma motion in and around them. Simulations and case study results suggest that jets also modify the magnetic field in the magnetosheath on their passage, aligning it more with their velocity. Based on MMS jet observations and corresponding superposed epoch analyses of the angles $\phi$ between the velocity and magnetic fields, we can confirm that this suggestion is correct. However, while the alignment is more significant for faster than for slower jets, and for jets observed close to the bow shock, the overall effect is small: Typically, reductions in $\phi$ of around $10°$ are observed at jet core regions, where the jets' velocities are largest. Furthermore, time series of angles $\phi$ pertaining to individual jets significantly deviate from the superposed epoch analysis results. They usually exhibit large variations over the entire range of $\phi$: $0°$ to $90°$. This variability is commonly somewhat larger within jets than outside, masking the systematic decrease in $\phi$ at core regions of individual jets.

## 1 Introduction

The region downstream of the Earth's bow shock, the magnetosheath, is oftentimes permeated by localized plasma entities of significantly enhanced dynamic pressure, so-called magnetosheath jets (for a recent review, see Plaschke et al., 2018). Within those jets, the dynamic pressure can easily exceed values measured in the pristine solar wind, and a significant fraction of jets even feature super-magnetosonic plasma velocities (Savin et al., 2008; Hietala et al., 2009; Plaschke et al., 2013; Savin et al., 2014). Thus, jets are highly distinctive phenomena in the subsolar magnetosheath.

Jets are known to occur more often downstream of the quasi-parallel shock (Archer and Horbury, 2013; Plaschke et al., 2013, 2016). In the subsolar magnetosheath, their occurrence is, hence, enhanced when the interplanetary magnetic field (IMF) points in a quasi-radial direction, i.e., when the angle between the IMF and the Earth-Sun-line – the IMF cone angle – is low. Under these conditions, shock-reflected particles are able to propagate along the IMF into the region upstream of the shock, where the particles then interact with the solar wind. The interaction region, called foreshock, exhibits localized magnetic field and plasma structures (e.g., short large amplitude magnetic structures, SLAMS) and waves that are convected back to

the shock, merge into it and, thus, continuously form and reform it (e.g., Schwartz and Burgess, 1991; Omidi et al., 2005; Blanco-Cano et al., 2006a, b). As a result, the quasi-parallel shock may be regarded as undulated or rippled. At the inclined surfaces of such ripples, solar wind plasma may be less decelerated and heated, yet still compressed and focused, yielding coherent high-speed jets within slower ambient plasma in the downstream magnetosheath region (Hietala et al., 2009, 2012).

As suggested by Karlsson et al. (2015, 2018) and shown in simulations by Palmroth et al. (2018), SLAMS themselves may become jets as they propagate through the undulated bow shock.

A second, smaller group of jets appears to be associated to the passage of IMF discontinuities, in particular when the character of the shock changes from quasi-perpendicular to quasi-parallel (Dmitriev and Suvorova, 2012; Savin et al., 2012; Archer et al., 2012; Plaschke et al., 2017). In this context, jets have also been associated to hot flow anomalies (HFAs) that can

occur when an IMF discontinuity interacts with the bow shock (Schwartz et al., 2000; Omidi and Sibeck, 2007).

Jets link the processes in the foreshock and at the bow shock with effects at the magnetopause, in the magnetosphere, and on the ground. Upon impact on the magnetopause, jets are able to indent the boundary significantly (e.g., Shue et al., 2009; Amata et al., 2011), launching waves on the surface of the magnetopause and in the magnetosphere (Plaschke and Glassmeier, 2011; Archer et al., 2013a, b; Archer et al., 2019), and/or triggering magnetic reconnection (Hietala et al., 2018). Effects of

the interaction are also visible from the ground, as ionospheric flow enhancements, geomagnetic variations, or dayside auroral activity (Hietala et al., 2012; Dmitriev and Suvorova, 2012; Han et al., 2016, 2017; Wang et al., 2018). Jets are very common in the magnetosheath. In general, large scale jets - larger than 2 Earth radii ($R_{\mathrm{E}}$) in diameter - hit the magnetopause approximately every 20 minutes. Under low IMF cone angle conditions, this rate increases to approximately one jet every 6 minutes (Plaschke et al., 2016). Note that typical jet scale sizes are on the order of $1\,R_{\mathrm{E}}$.

Recently, the inner structure of jets and their interaction with ambient magnetosheath plasma and fields have gotten more attention (Karimabadi et al., 2014; Plaschke et al., 2017; Plaschke and Hietala, 2018): When jets plough through slower ambient plasma, that latter plasma is pushed out of the way. Behind the jets, ambient plasma moves in to refill the wake. In addition, the fast motion of jets through slower ambient plasma may modify the magnetic field inside jets and in their vicinity, as seen in simulations by Karimabadi et al. (2014): The field may become more aligned with the plasma flow inside jets (see Figure 1a).

This hypothesis is supported by Plaschke et al. (2017), who found magnetic field and velocity measurements to be correlated within 18 jets that occurred during an hour-long interval. However, their case study could not yield conclusive evidence on how the magnetic field changes, on average, on the passage of a jet. The purpose of this paper is to obtain and present this information.

The results of this study are relevant in the context of solar wind - magnetosphere coupling, as the magnetosheath plasma

and fields represent the input to any interaction with the geomagnetic field at the magnetopause. Jet-induced changes in the magnetic field are expected to have repercussions on magnetosheath current sheets, on reconnection within the magnetosheath (Vörös et al., 2017) and at the magnetopause (Hietala et al., 2018), as well as on the associated triggering of substorms (Nykyri et al., 2019).

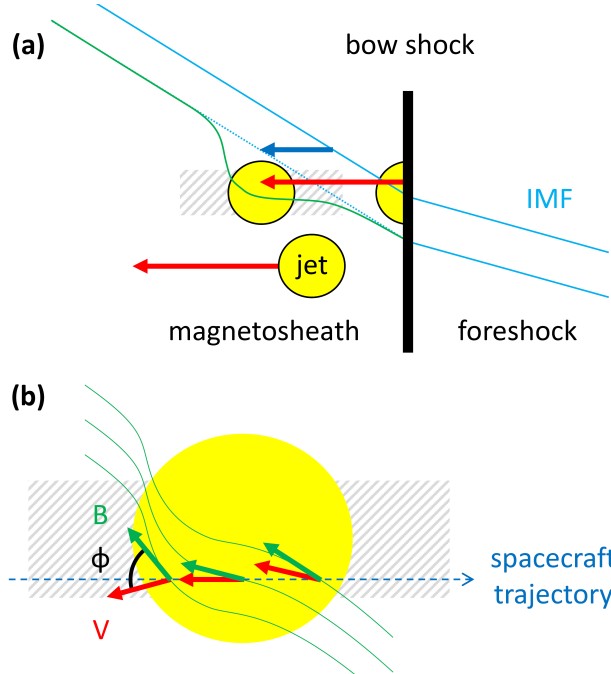

**Figure 1.** Top panel (a): Sketch of how magnetic fields in the magnetosheath may be modified by the motion of fast plasma jets. Velocities of jets and ambient plasmas are illustrated by red and blue arrows, respectively. In this paper, magnetic and velocity fields within the hatched area are evaluated. After Figure 12 in Plaschke et al. (2017). Bottom panel (b): Close-up on a jet. Green and red arrows show local directions of the magnetic field $B$ and velocity $V$ measured by a spacecraft on its trajectory through the jet. The angle between $B$ and $V$ is $\phi_{B,V}$.

## 2   Data and Methods

This study is based on jet observations by the four Magnetospheric Multiscale (MMS) spacecraft (Burch et al., 2016), made during the first and second dayside seasons of the mission (between 1 September 2015, the start of mission phase 1a, and 1 May 2017, the end of phase 2a). The MMS spacecraft were launched on 13 March 2015 into a highly elliptical, and nearly equatorial orbit. The initial apogee distance of the spacecraft from Earth was $12\,R_{\mathrm{E}}$. This distance stayed the same in 2015 and 2016, and was raised in the first few months of 2017 to follow the dawn magnetopause as the orbit swept westwards. Consequently, the spacecraft spent significant time in the vicinity of the subsolar magnetopause, flying in close tetrahedral configuration with spacecraft separations between $60$ and less than $10\,\mathrm{km}$, to achieve their primary goal: to investigate the small-scale physics of magnetic reconnection. While in the magnetosheath, they observed numerous jets.

To obtain a data set of jet observations by the MMS spacecraft, we follow the steps described in detail in Plaschke et al. (2013). We preselect intervals where the MMS spacecraft were located within a $30°$ wide cone centered at Earth and open to the Sun ($\sim 10$ to $14\,\mathrm{h}$ in local time), at distances above $7\,R_{\mathrm{E}}$ and below $18\,R_{\mathrm{E}}$ from the Earth's center. Within those preselected intervals, magnetosheath intervals are identified by the ion density surpassing twice the density in the solar wind. Here, we

use MMS ion density moments from the Fast Plasma Investigation (FPI, Pollock et al., 2016). These are compared to proton density measurements from NASA's OMNI high resolution data set (King and Papitashvili, 2005), averaged over 5 minutes preceding any time of interest. Note that OMNI measurements are based on solar wind monitor data from, e.g., the Advanced Composition Explorer (ACE) and Wind spacecraft, propagated to the bow shock nose. The 5 minute averaging accounts for

further propagation to the positions of the MMS spacecraft, closer to the magnetopause. In addition, within magnetosheath intervals the ion omni-directional energy flux density of $1\,\mathrm{keV}$ ions (measured also by FPI) shall be larger than that of $10\,\mathrm{keV}$ ions, to exclude magnetospheric observations. The magnetosheath intervals shall be at least 2 minutes long and all quantities of interest shall be available, i.e., magnetic field measurements by the MMS Fluxgate Magnetometers (FGM, Russell et al., 2016; Torbert et al., 2016), ion moments and distribution functions by FPI, and OMNI solar wind magnetic field and ion moments.

Therewith, MMS 1 to 4 yield a total of 4345.5 hours of magnetosheath data in 9375 intervals. Note that the intervals are almost equally distributed among the four MMS spacecraft, due to their close configuration: MMS 1, 2, 3, and 4, contribute 2376, 2370, 2279, and 2350 intervals, respectively.

Within these magnetosheath intervals, we search for jets as described in Plaschke et al. (2013). The main criterion is based on the dynamic pressure in the anti-sunward, i.e., $x$-direction in geocentric solar ecliptic (GSE) coordinates: $P_{\mathrm{dyn},x} = \rho V_x^2$.

Here $\rho$ is the ion (proton) mass density and $V_x$ the velocity in the $x$-direction. $P_{\mathrm{dyn},x}$ – measured by MMS – shall surpass half the pristine solar wind value, as determined from OMNI solar wind data ($P_{\mathrm{dyn},x} > P_{\mathrm{dyn,sw}}/2$). A jet interval is then defined by: $P_{\mathrm{dyn},x} > P_{\mathrm{dyn,sw}}/4$. One minute long intervals before the start and after the end of the jet intervals are denoted as pre-jet and post-jet intervals. All pre-jet, jet, and post-jet intervals shall be within one magnetosheath interval as defined above.

The times of maximum ratio of dynamic pressures $P_{\mathrm{dyn},x}/P_{\mathrm{dyn,sw}}$ (magnetosheath over solar wind) are denoted as $t_0$. We

require $V_x$ to be negative within jet intervals. $|V_x|$ should fall below half of its value at $t_0$ within both pre- and post-jet intervals, as specified in Plaschke et al. (2013). Applying all those criteria, we obtain a data set of 9757 jets, where MMS 1, 2, 3, and 4 contribute 2460, 2466, 2354, and 2477 jets, respectively. Obviously, due to the small spacecraft separations, jets seen by one spacecraft are likely to be seen by the other three spacecraft as well.

Similar to Plaschke and Hietala (2018), we introduce normalized times $t_\mathrm{n} = -2\ldots2$: $t_\mathrm{n} = -2$ corresponds to the start of the

pre-jet interval, $t_\mathrm{n} = -1$ is the start of the jet interval, $t_\mathrm{n} = 0$ equals $t_0$, i.e., the time of maximum dynamic pressure ratio in the jet core, $t_\mathrm{n} = 1$ denotes the end of the jet interval, and $t_\mathrm{n} = 2$ would be the end of the post-jet interval. Normalized times are defined for all 9757 jets. Note that normalized times $t_\mathrm{n} = -2\ldots2$ correspond with times $0\ldots4$ in Plaschke and Hietala (2018).

Figure 2 shows one of these jets, exemplarily, observed by MMS 1 on 24 December 2016. As can be seen in Figure 2c, the ion density clearly exceeded twice the corresponding solar wind values, indicating the presence of MMS 1 in the magnetosheath.

This is in agreement with Figure 2d showing the ion omni-directional energy flux density, also indicating MMS 1 to be immersed in thermalized magnetosheath plasma. Therein, the spacecraft observed a clear increase in GSE $V_x$ (Figure 2b), corresponding with a large increase in $P_{\mathrm{dyn},x}$ (Figure 2e) over the threshold of half the solar wind dynamic pressure. The vertical lines in the figure indicate the normalized times $t_\mathrm{n} = -2$ to 2, at 05:12:52, 05:13:52, 05:14:29, 05:15:23, and 05:16:23 UT, respectively. We can use these normalized times to perform superposed epoch analyses, based on pre-jet, jet, and post-jet

data: Therefore, the respective time intervals are compressed/expanded to become equal between integer $t_\mathrm{n}$.

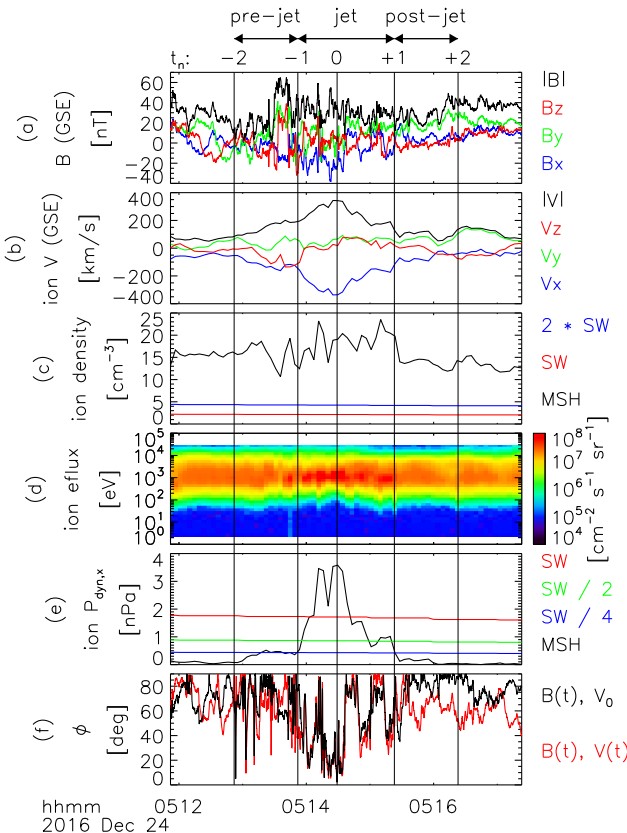

**Figure 2.** Jet example: MMS 1 magnetosheath and OMNI solar wind data of 24 December 2016. From top to bottom: (a) magnetic field $B$ in GSE, (b) ion velocity $V$ in GSE, (c) ion density in the magnetosheath in black and (twice) the ion density in the solar wind in red (blue), (d) magnetosheath ion energy flux density, (e) $P_{dyn,x}$ in the magnetosheath in black and in the solar wind in red (half and one quarter thereof in green and blue), and (f) angles $\phi_{B,V_0}$ in black and $\phi_{B,V}$ in red based on magnetosheath observations. Vertical lines show normalized times $t_n = -2$ to 2.

Note that time intervals between $t_n = -2$ and $-1$, and between $t_n = 1$ and 2 are 1 minute long by definition. The median lengths of time intervals between $t_n = -1$ and 0, and between $t_n = 0$ and 1 are 20 s (lower and upper quartiles: 10 s and 37 s) and 19 s (lower and upper quartiles: 10 s and 39 s), respectively. Hence, in "real" time, the jet interval length can vary significantly, while typically being one third as long as the pre and post-jet intervals combined (see also Plaschke et al., 2013).

Finally, we determine the relative locations $r_{rel}$ of jet-observing spacecraft at times $t_n = 0$ between the magnetopause ($r_{rel} = 0$) and the bow shock ($r_{rel} = 1$). Therefore, we use the magnetopause and bow shock models by Shue et al. (1998) and Merka et al. (2005), respectively (see, Plaschke et al., 2013; Hietala and Plaschke, 2013). OMNI solar wind data pertaining to jet times are the input conditions to the model calculations. There are 1856 jets observed closest to the magnetopause ($r_{rel} < 0.25$) and 797 jet observed closest to the bow shock ($r_{rel} > 0.75$). Hence, the vast majority of jets are associated to central locations within the subsolar magnetosheath.

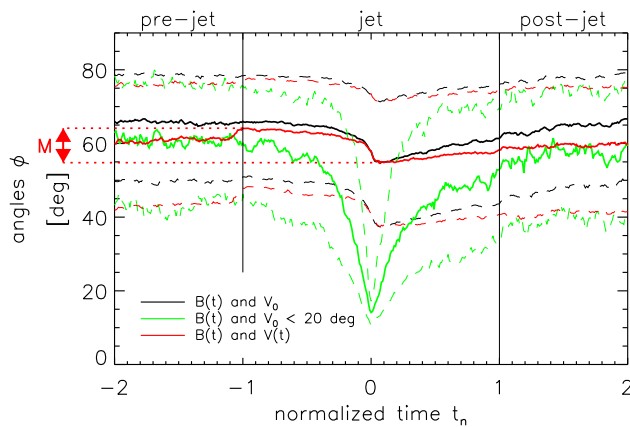

**Figure 3.** Superposed epoch analyses of the angles $\phi_{B,V}$ in red, $\phi_{B,V_0}$ in black, and $\phi_{B,V_0}$ of those jets where that angle is limited to $20°$ at $t_n = 0$ in green. Solid lines show median values, dashed lines show upper and lower quartile values. Red dotted lines mark minimum and maximum values of median $\phi_{B,V}$ angles: the difference between these two values is $M_{B,V} = 9.4°$.

## 3    Results

The primary objective of this paper is to show whether (or not) the magnetic field aligns with the flow velocity on jet passage, as suggested by simulation results presented in Karimabadi et al. (2014) and case study observations by Plaschke et al. (2017). This can be answered by a superposed epoch analysis of the angle $\phi_{B,V}$ (see Figure 1b) between magnetic field $B(t)$ and ion

velocity $V(t)$ vectors. The result is shown in red in Figure 3 (see also Figure 2f, red line, for a contributing example). The solid line shows median values and the dashed lines illustrate the upper and lower quartiles. Note that the angles $\phi$ in all figures are acute angles, i.e., restricted between $0°$ and $90°$. We have checked that this does not limit the angular deflections resulting from the superposed epoch analyses.

   Let's focus first on the edges of the jet interval. Before and after that interval, in the pre- and post-jet intervals, the angle

$\phi_{B,V}$ is approximately $60°$ and constant. At $t_n = -1$, a slight increase in the median and lower quartile of $\phi_{B,V}$ can be seen. This corresponds to the increase in dynamic pressure $P_{\mathrm{dyn},x}$ over one quarter of the solar wind value. At $t_n = 1$, the end of the jet interval, no significant feature in $\phi_{B,V}$ can be discerned. Instead, at that time, $\phi_{B,V}$ is gradually recovering from a decrease that sharply happens at $t_n = 0$.

   The normalized time $t_n = 0$ (or $t_0$) is of special importance, as it marks the time of maximum dynamic pressure in the jet,

the jet core. Decreases in $\phi_{B,V}$ at that time show that, generally, there is "some" alignment of $B$ and $V$ happening inside jets. However, in the superposed epoch analysis, this effect is limited: The difference $M_{B,V}$ between the maximum and the minimum of the median angle $\phi_{B,V}$ is $9.4°$. $M_{B,V}$ is indicated by a red arrow in Figure 3.

   Angles $\phi_{B,V_0}$ can also be computed by using the velocity vector at that specific time ($V_0 = V(t_0)$) and by comparing it with time series of magnetic field vectors $B(t)$. The direction of $V_0$ should be a good indication of the overall jet propagation

direction. Note that good deHoffmann-Teller frames exist for almost all jets, and that the directions of $V_0$ are generally

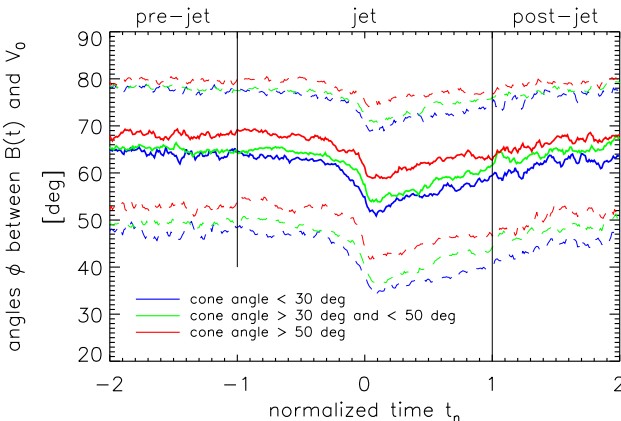

**Figure 4.** Superposed epoch analyses of the angle $\phi_{B,V_0}$, using only jets occurring under IMF cone angles of $< 30°$ (blue, 2811 contributing jets), between $30°$ and $50°$ (green, 4119 contributing jets), and above $50°$ (red, 2827 contributing jets), respectively. As in Figure 3, solid lines show median values and dashed lines show upper and lower quartiles.

consistent with the directions of the deHoffmann-Teller frame velocities, computed from $V$ and $B$ measurements between normalized times $t_n = -1$ and 1 (Sonnerup et al., 1990).

Results of the superposed epoch analysis of $\phi_{B,V_0}$ are shown in black in Figure 3 (see also Figure 2f, black line, for a contributing example). In this case, the median $\phi_{B,V_0}$ shows no variation at $t_n = -1$. The decrease at $t_n = 0$ is a bit deeper
$(M_{B,V_0} = 12.1°)$, because the overall value of $\phi_{B,V_0}$ within the pre- and post-jet intervals is slightly higher, approximately at $65°$.

The limited alignment effect apparent at $t_n = 0$ raises the question whether the considered effect is significant in any of the jets. Therefore, we select those jets where $\phi_{B,V_0} < 20°$ at $t_n = 0$. This holds for 449 jets, i.e., for $4.6\%$ of the jet data set. Note that the example jet shown in Figure 2 belongs to this group. The corresponding superposed epoch analysis of $\phi_{B,V_0}$ based
only on these jets is shown in green in Figure 3. Apparently, a major alignment of $B$ and $V$ does happen sometimes, although only in a small minority of cases. For this subsample of jets, $M_{B,V_0} = 49.6°$ is obtained.

The alignment effect may depend on the upstream solar wind or jet intrinsic conditions. As reported in Plaschke et al. (2013), the jet occurrence in the subsolar magnetosheath is heavily dependent on the IMF cone angle. The decrease in $\phi_{B,V_0}$ at $t_0$, however, is only weakly dependent on this quantity, as can be seen in Figure 4. In this figure, blue, green, and red
solid lines correspond to the median angles $\phi_{B,V_0}$ based on jets observed during low, medium, and high IMF cone angle conditions: $< 30°$, $30°$ to $50°$, and $> 50°$. Median cone angles associated to these categories are: $20.9°$, $39.8°$, and $61.3°$. The corresponding alignment effect strengths $M_{B,V_0}$ are $14.6°$, $13.9°$, and $10.5°$, respectively.

The overall $\phi_{B,V_0}$ levels also change slightly with the IMF cone angle, $B$ and $V$ being a few degrees more aligned, in general, under low IMF cone angle conditions. The same results with respect to cone angle dependence holds for the angles
$\phi_{B,V}$ as a function of $t_n$ (not shown). Note that using IMF cone angle measurements 20 minutes before $t_n = 0$ instead of at $t_n = 0$ noticeably increases the alignment effect strength for low cone angle events to $M_{B,V_0} = 17.3°$.

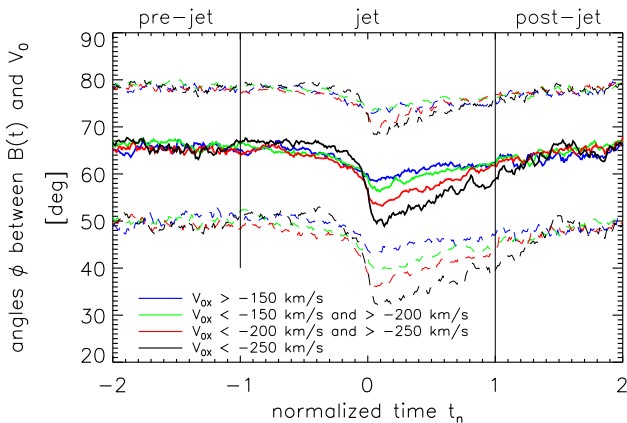

**Figure 5.** Superposed epoch analyses of the angle $\phi_{\boldsymbol{B},\boldsymbol{V}_0}$, using only jets featuring $V_{0x}(t_0) > -150\,\mathrm{km/s}$ in blue (1623 contributing jets), $-150\,\mathrm{km/s} > V_{0x}(t_0) > -200\,\mathrm{km/s}$ in green (3087 contributing jets), $-200\,\mathrm{km/s} > V_{0x}(t_0) > -250\,\mathrm{km/s}$ in red (2699 contributing jets), and $-250\,\mathrm{km/s} > V_{0x}(t_0)$ in black (2348 contributing jets). As in Figure 3, solid lines show median values and dashed lines show upper and lower quartiles.

The decrease in $\phi_{\boldsymbol{B},\boldsymbol{V}_0}$ at $t_0$ is more strongly dependent on the velocity of the jets (Figure 5). The larger the velocity at $t_0$ is, the larger the decrease will usually be in $\phi_{\boldsymbol{B},\boldsymbol{V}_0}$. Figure 5 shows superposed epoch analyses of this quantity as a function of $V_{0x}$ at $t_0$. The blue, green, red, and black solid lines correspond to the median angles $\phi_{\boldsymbol{B},\boldsymbol{V}_0}$ based on jets featuring velocities $V_{0x} > -150\,\mathrm{km/s}$, $-200$ to $-150\,\mathrm{km/s}$, $-250$ to $-200\,\mathrm{km/s}$, and $< -250\,\mathrm{km/s}$. The median velocities $V_{0x}$ associated to

these four categories are: $-130\,\mathrm{km/s}$, $-175\,\mathrm{km/s}$, $-221\,\mathrm{km/s}$, and $-293\,\mathrm{km/s}$. The corresponding alignment effect strengths $M_{\boldsymbol{B},\boldsymbol{V}_0}$ are in these cases $8.9°$, $11.3°$, $14.7°$, and $18.8°$, respectively. There is a clear linear dependency of $M_{\boldsymbol{B},\boldsymbol{V}_0}$ on the median $V_{0x}$ values of the form: $M_{\boldsymbol{B},\boldsymbol{V}_0} = 0.8669° - (0.0612°\,\mathrm{s/km})\,V_{0x}$.

Finally, we check the change in $\phi_{\boldsymbol{B},\boldsymbol{V}_0}$ on jet passage as a function of the location of observation between the magnetopause ($r_{\mathrm{rel}} = 0$) and the bow shock ($r_{\mathrm{rel}} = 1$). The results of the corresponding superposed epoch analyses are shown in Figure 6.

As can be seen, the green and red traces corresponding to mid-sheath jets ($0.25 < r_{\mathrm{rel}} < 0.75$) are almost identical to each other and also extremely similar to the black line in Figure 3. There are, however, deviations in the alignment of the magnetic and velocity fields when it comes to jets observed closest to the magnetopause ($r_{\mathrm{rel}} < 0.25$, blue line) and closest to the bow shock ($r_{\mathrm{rel}} > 0.25$, black line). In the former case, $M_{\boldsymbol{B},\boldsymbol{V}_0} = 11.6°$ is not dissimilar to the the overall value of $12.1°$, but the alignment effect seems less concentrated around $t_{\mathrm{n}} = 0$. In the latter case, the alignment effect is clearly stronger and we obtain

$M_{\boldsymbol{B},\boldsymbol{V}_0} = 21.1°$.

It should be noted that the MMS spacecraft are more likely to observe the bow shock when the entire magnetospheric system is compressed, i.e., when the solar wind dynamic pressure $P_{\mathrm{dyn,sw}}$ is high. In agreement therewith, the mean $P_{\mathrm{dyn,sw}}$ values pertaining to the four categories $r_{\mathrm{rel}} < 0.25$, between $0.25$ and $0.5$, between $0.5$ and $0.75$, and $r_{\mathrm{rel}} > 0.75$ are: $P_{\mathrm{dyn,sw}} = 1.77\,\mathrm{nPa}$, $2.22\,\mathrm{nPa}$, $2.74\,\mathrm{nPa}$, and $3.39\,\mathrm{nPa}$, respectively. This raises the question whether the alignment effect is strongly

dependent on the upstream dynamic pressure. The answer to this question is displayed in Figure 7.

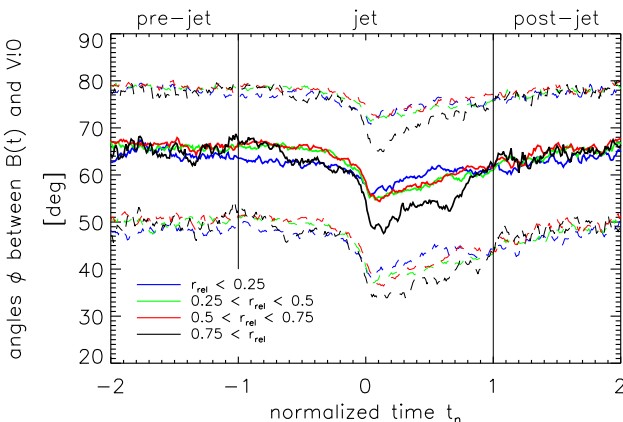

**Figure 6.** Superposed epoch analyses of the angle $\phi_{B,V_0}$ as a function of the relative location of jet observations between the magnetopause ($r_{\rm rel} = 0$) and the bow shock ($r_{\rm rel} = 1$). As in Figure 3, solid lines show median values and dashed lines show upper and lower quartiles.

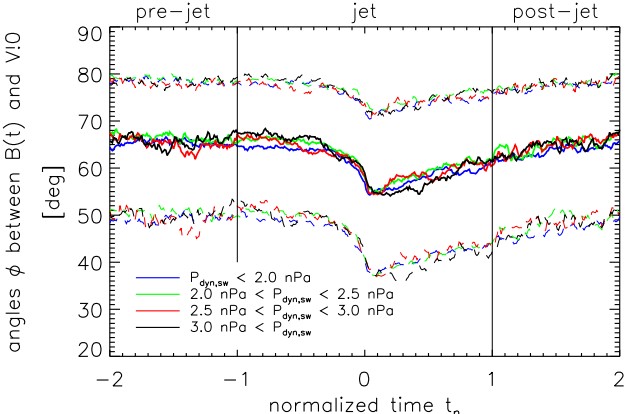

**Figure 7.** Superposed epoch analyses of the angle $\phi_{B,V_0}$ as a function of the upstream solar wind dynamic pressure $P_{\rm dyn,sw}$. As in Figure 3, solid lines show median values and dashed lines show upper and lower quartiles.

As can be seen in that figure, higher $P_{\rm dyn,sw}$ values are not associated with significant increases in alignment between $B$ and $V$ at $t_{\rm n} = 0$. We have also tested the relation of other upstream solar wind conditions (velocity, density, magnetic field strength, and Mach numbers) to the timeseries of angles $\phi_{B,V_0}$ and $\phi_{B,V}$. We have not found any indications of these conditions being related to larger systematic changes in alignment.

## 4  Discussion

The typical angles between magnetic field and plasma flow directions in the subsolar magnetosheath are reflected at normalized times $t_{\rm n} = -2$ and 2, at the ends of the superposed epoch analyses. As shown in Figures 3 to 7, the median angles $\phi_{B,V}$ and

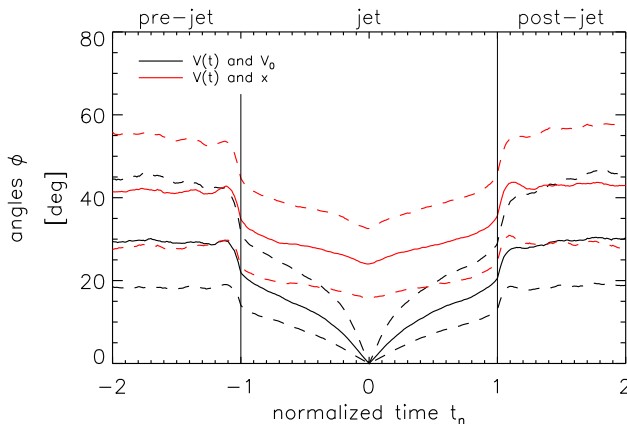

**Figure 8.** Superposed epoch analysis of the angle $\phi_{\boldsymbol{V},\boldsymbol{V}_0}$ between $\boldsymbol{V}(t)$, the time series of velocity vectors, and $\boldsymbol{V}_0$, the vectors at times $t_0$ in black. In red, the superposed epoch analysis of $\phi_{\boldsymbol{V},\boldsymbol{e}_x}$ is shown, where $\boldsymbol{e}_x$ is the unit vector in GSE $x$-direction.

$\phi_{\boldsymbol{B},\boldsymbol{V}_0}$ at these times are found to be between approximately $60°$ and $70°$. At first glance, such high values seem remarkable, taking into account that they are also found under low IMF cone angle conditions (blue line in Figure 4). However, they may be explained to a great extent by typical draping of the IMF in the magnetosheath. The median angle $\phi_{\boldsymbol{B},\boldsymbol{V}}$ of all magnetosheath observations by the MMS spacecraft selected for this study is $59.2°$. This value corresponds quite well with median angles

$\phi_{\boldsymbol{B},\boldsymbol{V}}$ at times $t_\mathrm{n} = -2$ and $2$ (red solid line in Figure 3). Note, however, that this angle is specific to the distribution of locations of the MMS spacecraft in the subsolar magnetosheath. Different locations, e.g., towards the flanks, will be associated to different typical angles $\phi_{\boldsymbol{B},\boldsymbol{V}}$, which are a function of the combined draping and flow patterns.

The first jet-induced deviations in $\phi_{\boldsymbol{B},\boldsymbol{V}}$ and $\phi_{\boldsymbol{B},\boldsymbol{V}_0}$ are seen at $t_\mathrm{n} = -1$. At this time, the median angle $\phi_{\boldsymbol{B},\boldsymbol{V}}$ increases slightly, while $\phi_{\boldsymbol{B},\boldsymbol{V}_0}$ does not change. As $\boldsymbol{V}_0$ stays constant, the change necessarily has to come from a change in $\boldsymbol{V}$ at

$t_\mathrm{n} = -1$. This change is reflected in Figure 8, which shows superposed epoch analyses of the angles between $\boldsymbol{V}(t)$ with $\boldsymbol{V}_0$ and $\boldsymbol{e}_x$ in black and red, respectively. Here, $\boldsymbol{e}_x$ is the unit vector in GSE $x$-direction, along the Earth-Sun-line.

As can be seen in the figure, between $t_\mathrm{n} = -1$ and $1$ the jet-related plasma deflection takes place, with jets propagating more in the anti-sunward direction than the ambient magnetosheath plasma. This feature is typical for jets and has been reported, e.g., by Karlsson et al. (2012), Archer and Horbury (2013), and Plaschke et al. (2013). Apparently, the flow deflection does not

affect the magnetic field direction, so that $\phi_{\boldsymbol{B},\boldsymbol{V}_0}$ stays constant at $t_\mathrm{n} = -1$. After that time, $\boldsymbol{V}$ gradually approaches $\boldsymbol{V}_0$, as reflected in Figure 8 (see black line). Consequently, angles $\phi_{\boldsymbol{B},\boldsymbol{V}}$ and $\phi_{\boldsymbol{B},\boldsymbol{V}_0}$ behave rather similarly close to $t_\mathrm{n} = 0$. This can also be seen in Figure 2f, showing black and red lines closely aligned at $t_\mathrm{n} = 0$ but deviating more strongly before $t_\mathrm{n} = -1$ and, in particular, after $t_\mathrm{n} = 1$.

In light of the decreases of $\phi_{\boldsymbol{B},\boldsymbol{V}_0}$ at $t_\mathrm{n} = 0$, we can confirm that jets modify magnetic fields in the magnetosheath, tending

to align them with their direction of propagation. This alignment happens sharply at $t_\mathrm{n} = 0$, i.e., at the cores of the jets that feature the fastest plasma (see Plaschke and Hietala, 2018). However, it is also clear from the statistics presented in this paper, that the alignment effect is generally small - much smaller than seen in simulations by Karimabadi et al. (2014). The reason

for this discrepancy might be the restrictions imposed on plasma motion in their simulations, as they were 2-dimensional (2D) and not 3D.

In general, median $\phi_{B,V_0}$ angles decrease by approximately $M_{B,V_0} \approx 10°$. The statistics including only the fastest jets exhibit a decrease $M_{B,V_0}$ by approximately $20°$, and so do the statistics including only jets observed close to the bow shock ($r_{\rm rel} > 0.75$). The fact that faster jets lead to a stronger alignment of $B$ and $V$ is not surprising, as the velocity difference between jets and ambient plasmas should be responsible for the change in magnetic field direction (see Figure 1a). As jets plough through slower plasma, they should drag the frozen-in magnetic field with them, straightening it at and after their passage (Plaschke et al., 2017). This picture is also in agreement with the gradual recovery of $\phi_{B,V_0}$ after the passage of the jet core, starting at $t_{\rm n} = 0$ and extending beyond $t_{\rm n} = 1$.

The fact that the alignment effect of $B$ and $V$ is stronger for jets observed close to the bow shock (the source region) is, however, somewhat puzzling. As a consequence, it can hardly be argued that the alignment effect increases as jets progress through the magnetosheath towards the magnetopause. Instead, the alignment may decrease as jets evolve. This may be due to the boundary conditions imposed by the magnetopause. The composition of jets observed close to the bow shock and the magnetopause may also be different. As reported in Plaschke et al. (2013), relatively more jets are observed close to the bow shock than close to the magnetopause. Hence, only a certain fraction of jets makes it all the way through the magnetosheath. It cannot be excluded that the alignment effect is generally smaller for that subset of jets.

A relatively large angular deviation of $M_{B,V_0} = 17.3°$ is also obtained for jets that were launched into a low IMF cone angle magnetosheath (cone angle $< 30°$ 20 minutes before $t_{\rm n} = 0$). This result may suggest that the condition or state of the magnetosheath prior to jet generation may also have an influence on the alignment effect in particular and on jet evolution in general.

It shall be noted that all the results presented here pertain to changes in $B$ and $V$ emerging from superposed epoch analyses of thousands of jets. Individual jets can and will look very different. As shown in Figure 3 in green, there are jets ($< 5\%$) featuring quite small $\phi_{B,V_0} < 20°$ at $t_0$. The example jet shown in Figure 2 is one of them. As can be seen in the bottom panel of Figure 2, $\phi_{B,V_0}$ changes a lot over the passage of this particular jet, which is not special in this respect. Within its jet interval, between $t_{\rm n} = -1$ and $t_{\rm n} = 1$, $\phi_{B,V_0}$ values close to $0°$ and $90°$ are reached in rapid succession.

To quantify this variability statistically, we compute the inter-quartile range $\sigma$ of $\phi_{B,V}$ within $10\,{\rm s}$-wide sliding time intervals for every jet. The corresponding superposed epoch analysis of $\sigma(\phi)$ is presented in Figure 9. Variability on the order of $14°$ seems to be typical. The median variability slightly increases within jet intervals to about $17°$ at $t_{\rm n} = 0$. This increase is also suggested by the example displayed in Figure 2. Note that the variability in $\phi_{B,V}$ is of the same order as the typical alignment at $t_0$, quantitatively supporting the observation that the alignment is hard to discern in individual events.

## 5   Conclusions

The purpose of this paper is to ascertain whether the high-speed motion of magnetosheath jets through slower ambient plasma leads to an alignment of magnetic and velocity fields, as predicted by simulations (Karimabadi et al., 2014) and case study

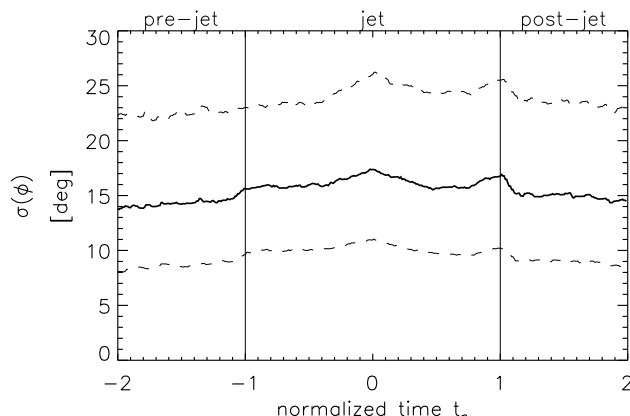

**Figure 9.** Superposed epoch analysis of the inter-quartile range of angles $\phi_{B,V}$ within 10 s-wide time intervals, centered around respective normalized times. Solid line depicts median, dashed lines upper and lower quartiles.

observations (Plaschke et al., 2017). To address this question, we have performed superposed epoch analyses of the angles $\phi_{B,V}$ and $\phi_{B,V_0}$ as a function of normalized times $t_n$, based on MMS jet observations in the subsolar magnetosheath. These are our main results:

– In agreement with expectations, jets generally do modify the magnetic field on their passage, aligning it more with their velocity. This alignment takes place at the core of the jets, at $t_0$, and it is significantly stronger for faster jets and for jets observed close to the bow shock. Recovery to usual angles $\phi$ occurs gradually within the trailing part of the jets.

– The alignment effect is not (strongly) dependent on the IMF cone angle, IMF strength, solar wind velocity, density, dynamic pressure, or Mach numbers.

– In disagreement with simulations by Karimabadi et al. (2014), this alignment is relatively small. Typically, the angles $\phi$ change only by about $10°$. The reason for this discrepancy might be the restrictions imposed on plasma motion in the simulations, as they are 2D and not 3D.

– Time series of $\phi$ of individual jets look very different to the superposed epoch analysis results: Large fluctuations in $\phi$ on sub-jet time scales are very common. This variability is somewhat larger within jets than outside, masking the decrease in $\phi$ at times $t_0$ of individual jets.

*Data availability.* The FGM and FPI data used in this paper are stored at the MMS Science Data Center (https://lasp.colorado.edu/mms/sdc/) and are publicly available. The OMNI solar wind data are publicly available from the NASA Space Physics Data Facility at the Goddard Space Flight Center (https://omniweb.gsfc.nasa.gov/ow_min.html).

*Author contributions.* FP conceived the study and MJ did a significant part of the data analysis work. HH and LV helped with the discussion and interpretation of the results.

*Competing interests.* The authors declare that no competing interests are present.

*Acknowledgements.* The dedication and expertise of the Magnetopheric MultiScale (MMS) development and operations teams are greatly appreciated. We acknowledge the use of Level 2 fast survey Flux-Gate Magnetometer (FGM) and Fast Plasma Investigation (FPI) data. We acknowledge valuable discussions within the International Space Science Institute (ISSI) team called "Jets downstream of collisionless shocks" led by two authors of this paper (FP and HH). The work at the University of Turku was supported by the Turku Collegium of Science and Medicine. The work of HH was supported by National Aeronautics and Space Administration (NASA) grant NNX17AI45G, NASA contract NAS5-02099, and the Royal Society University Research Fellowship URF\R1\180671.

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
