# Peer review of "On the alignment of velocity and magnetic fields within magnetosheath jets"

_Annales Geophysicae, 2019_

## Referee Comment (RC1) · Anonymous Referee #1 · 2 Aug 2019

The manuscript presents statistical results of how the angle between the local magnetic field and velocity vectors varies during magnetosheath jet events. This work has been motivated by recent simulations and case study observations. Given the myriad of impacts on Earth's magnetosphere that magnetosheath jets can have, understanding their propagation from their bow shock origin to the magnetopause is important and has largely been an open question in this particular topic within the solar wind - magnetosphere dynamical coupling. The methodology and results are clear and well presented and the results quantitatively align with previous theoretical interpretations of previous work, lending the statistical weight to these. I recommend publication subject to the authors addressing a number of minor issues.

General Comments:

[Figure]

It was not clear to the reviewer whether the angles used (phi) were limited to be the acute angles (0-90 degrees) between the vectors. The figure limits throughout suggest this may be the case. However, while perhaps unlikely, it might be possible under certain configurations that a jet could bend the magnetic field lines back on themselves significantly resulting in angular deflections greater than 90 degrees which this analysis would not capture. This would result in the wrong angle being measured in the deflected regions. The authors should check that no greater than 90 degree deflections within the jet from pre/post occur in the dataset. If they do, the authors will need to re-do the analysis using the full angle between the vectors. They may wish to counteract the effects of the sign of Bx, which would lead to two separate populations in the data corresponding to either side of the heliospheric current sheet, by the average pre/post interval sign of Bx into account when calculating the angles for each event.

In several cases, the authors quote median values as well as standard deviations. However, commonly a standard deviation is a difference from the mean value rather than the median. Medians are appropriate here as the mean is likely to be affected by outliers. So to would even a standard deviation about the median be affected. Quoting the lower and upper quartiles would be more appropriate throughout.

The abstract did not make it clear that there is a significant trend in deflection angles with jet speed. This is a key result of the paper and should be made more prominent in the abstract.

Specific Comments:

Page 1 Line 2 - The authors should also briefly comment on other scenarios such as that proposed by Karlsson et al. [2018, Ann. Geophys., https://doi.org/10.5194/angeo-36-655-2018] concerning SLAMS transmission through bow shock ripples which have recently been shown in Vlasiator simulations [Palmroth et al. [2018, Ann Geophys, https://doi.org/10.5194/angeo-36-1171-2018].

Page 4 Line 14 - Do the authors have an estimate on the number of independent jets

observed, taking into account those that were observed by the same spacecraft?

Page 4 Line 23 - It is unclear why twice the solar wind density is used as a comparison measure when the bow shock typically compresses the solar wind density by a factor 4. If the authors mean the enhancement in density was greater than twice the solar wind density they should so state and make this clearer in how this result is depicted in Figure 2.

Page 6 Line 17 - While the pre/post interval angle angle does decrease with cone angle (a result of the different draping patterns), it does appear that there is a very slight difference in the depth of the median deflections with cone angle. The authors should estimate these depths (the effect size) and the significance of any differences with cone angle more thoroughly.

Page 8 Line 1-2 - Can the authors comment more on expected draping angles at MMS's location for jet events or at least cite previous statistical studies into IMF draping near the magnetopause?

Page 8 Line 9 - The jet identification method does not necessarily mean a deflection towards the Sun-Earth line. Given the criteria, it could possibly have been the case that the y and z components of V similarly increased as Vx does which would not result in a deflection.

Page 8 Line 20 - The authors should perform simple estimates of expectations given the picture in Figure 1 i.e. from a purely geometric point of view, ignoring any resistive forces, how much deflection would be expected for the set of observed jets and draping angles purely by the jet's flow locally advecting the field lines. To what extent could e.g. magentic tension forces slow the jet's motion thereby reducing the deflection etc. This would bring into context the results and interpretation more clearly.

Page 9 Line 9 - The authors should also mention another statistical method which might be adopted - computing individual deflection depths based on the (average of) pre/post

jet intervals and estimating the distribution of deflection angles from this, rather than distributions of absolute angles. I am not advocating this be done for this manuscript, merely suggested as future work.

Technical Corrections:

Page 4 Line 12 - The statement "We require Vx to be negative within jet intervals and surpass half of its value at t0 within both pre- and post-jet intervals" is a little confusing and I would suggest the authors instead of discussing a negative number surpassing a threshold in the pre/post interval, instead talk about the absolute value dropping below said threshold.

Page 5 Line 4-5 - Please make it clear that values at each individual time are used, showing the same symbols as in the figures i.e. B(t) V(t). This will help contrast later with the other angles used.

Page 6 Line 6 - Please make clear that the magnetic field here is still taken at each individual time B(t).
* * *

---

## Referee Comment (RC2) · Anonymous Referee #2 · 7 Aug 2019

This paper presents the effects of magnetosheath jets on magnetosheath magnetic fields based on epoch analysis. The results clearly show that the magnetic fields tend to be aligned with the velocity of magnetosheath jets. The authors also discuss the consistency and inconsistency with the previous case studies and simulations. That discussion looks good to me. Thus I only have some minor comments as shown below.

Line 6 of Page 6: You use the angle between B(t) and V0 to do epoch analysis but can you explain why you did this? The angle between B(t) and V(t) is easy to understand while B(t) and V0 usually occur at different time/location, so it is difficult for me understand why you compare these two vectors. In addition, in figure 3, the alignment effect is more significant shown in black lines than red lines. Do you have a good explanation of that?

[Figure]

Figure 3 or Lines 1-3 of Page 7: I saw you discussed about why the angle between B and V is ∼60-70 degree at t(-2) or t(2). You said that this value indicates "the typical angles between magnetic field and plasma flow directions in the subsolar magnetosheath". Is there any reference showing that typical angle? In addition, do you think the locations of MMS probes (e.g., closer to magnetopause vs. closer to bow shock) affect that angle in the background magnetosheath plasma? Furthermore, magnetosheath jets may evolve in the magnetosheath as they propagate from bow shock to magnetopause, do the locations of MMS probes also affect the angle change with magnetosheath jets? Is it possible to briefly discuss about that with your current database?

Lines 16-17 of Page 8: "... much smaller than seen in simulations by Karimabadi et al., (2014)". You attributed it to the 2D not 3D simulation in the previous simulation. Their simulation seems to be done in the XY plane and do you agree that if you do an epoch analysis in that 2D plane, you will obtain the similar result as what their simulation obtained?

Lines 4-9 of Page 10: The second conclusion says the statistical results got smaller angle change than the previous simulation got; The third conclusion says that the large fluctuations in sub-jets may mask the decrease in âĹĚ. If there is a way to remove the effects by sub-jets (you don't have to do that), do you think the decrease in âĹĚ will be comparable to what the previous simulation shows? Or you still consider 2D vs. 3D is a important issue here?

---

## Referee Comment (RC3) · Anonymous Referee #3 · 8 Aug 2019

GENERAL COMMENTS:

This paper is well written, and presents the results of a large statistical study of the alignment of the flow direction associated with a magnetosheath high speed jet and the local magnetic field in a clear and concise fashion. The authors show that while there is a deviation of the local magnetic field direction such that it becomes more aligned with plasma flow direction of the jet, this is not a large effect on a statistical level, contrary to recent modeling results. While obtaining a deeper understanding of the general nature of the structure of the jets is important, there is not clear connection in the text as to why this particular aspect of the jet is important for further understanding local processes in the magnetosheath or how the alignment of the magnetic field and velocity vector impacts interactions of the jet with local ambient plasma and the magnetopause.

[Figure]

SPECIFIC COMMENTS:

Page 3: The "30°-wide cone centered at Earth and open to the Sun" used for selecting jet intervals simply corresponds to MLTs of 11-13 hours, correct?

Page 6, Line 15: Along with intrinsic conditions or upstream solar wind possibly contributing to the limited alignment effect, does spacecraft trajectory through the jet structure have any effect on the observations of the jet?

Page 6, Lines 16 – 18: Looking at Figure 4, there does appear to be a slight dependence on IMF cone angle. When you look at the percent change relative to the phi_B,V0 level at t = -2 for the different cone angle bins (i.e., looking at the change in the angle at t = 0 after subtracting out as an offset the value of the angle at t = -2 for each cone angle bin), is any dependence of cone angle seen?

Page 6, Lines 10-14 and Page 9, Line 6: Can you show another jet example that has the more common feature of a smaller change in the alignment of B and V? Since solar wind conditions were used for this statistical study, are there any indications that other upstream conditions may be related to the largest changes in alignment of B and V?

In the discussion section, more results are presented on the change in plasma velocity and the standard deviation of the angle between B and V, which is helpful in interpreting the superposed epoch analyses of the changes in alignment between B and V in the core of the jets. However, there isn't much discussion on the implications and consequences of the limited alignment effect seen for the majority of the jets. What does the small change in alignment mean for interactions with the local plasma or the magnetopause? Does it particularly matter, and if so, why? More discussion on this would be beneficial for grounding the results in the broader context of the studies mentioned in the introduction section.

TECHNICAL CORRECTIONS:

Page 2, line 8: change "on ground" to "on the ground" Page 2, line 9: add "of the

magnetopause" after "surface" Page 8, line 1: change "extend" to "extent" Page 8, line 8: change "in anti-sunward" to "in the anti-sunward" Page 8, line 12: change "similar" to "similarly"

---

## Referee Comment (RC4) · Anonymous Referee #4 · 3 Sep 2019

Overview:

Paper focuses on studying if there is statistical, observational, evidence supporting the simulations of Karimabadi et al., 2014 that magnetosheath jets make the ambient magnetic field more aligned with the jet velocity. Study uses data (FPI for ion moments and FGM for the magnetic field) from the four Magnetosphere Multi-Scale (MMS) spacecraft between September 2015 and May 2017. The main conclusions by the authors, obtained based on superposed epoch analysis of the pre and post jet angles between ion velocity and magnetic field, is that while jets generally modify the magnetic field, the alignment of the magnetic field with the jet flow is relatively small. They suspect the discrepancy may be due to the 2-D simulation geometry, while the real nature is 3-D.

Paper addresses a compelling topic as understanding of the jet formation, structure

and propagation thorough magnetosheath and their subsequent impact on the magnetopause will help address magnetospheric response to the dynamic driving by the solar wind and may be even relevant for the long-standing "internal" vs "external" substorm-triggering debate. The authors have compiled an extensive data base for addressing this topic using MMS data and I think the paper is suitable for publication after some revisions. I also think that the manuscript would strongly benefit from some additional analysis before clear conclusions can be made. I would recommend the authors to perform some additional analysis (which should not take too long) and address the following in the revised manuscript:

Main comments:

1. The superposed epoch analysis uses normalized time and organizes the data on pre- and post jet intervals based on this time. However, the number of single spacecraft measurements do not provide information on the 3-D jet structure, magnetosheath structure or how the spacecraft might have crossed the jet. Because the spacecraft are nearly stationary when compared to jet propagation speed, the "pre jet" and "post jet" time intervals may correspond to the vastly different spatial regions with respect to jets due to different 3-D field geometries that can arise due various factors, e.g., how spacecraft crosses the jet, the distance to the jet formation region (is the jet accelerating, moving at constant speed or decelerating), how does the jet dynamic pressure relate to the ambient magnetic field pressure in the magnetosheath, to the distance to the magnetopause and due to different spacecraft z-component in GSM coordinates. Note that during 2015-2017, close to the spring and fall equinox times, the MMS GSM z-coordinate in the dayside magnetosheath can be substantial so this list will likely include several high-latitude magnetosheath observations close the dayside high-latitude magnetopause.

I would recommend the authors perform and address the following in the revised manuscript:

a) Sort all of the identified jet intervals based on the distance to the model magnetopause and model bow shock calculated using prevailing solar wind condition during the jet observations.

b) Show a distribution of the MMS z-position during jet observations and study the dependence of the MMS z-coordinate and distance to the magnetopause and bow shock on the deviation of the "pre jet" and "post jet" angles. Study how the ratio of the local dynamic jet pressure and the pre-jet magnetic field pressure varies as function of distance between magnetopause and bow shock and how this affects the angles.

c) The analysis uses the velocity vector as measured by the MMS to calculate the angles for pre-jet, jet and post jet intervals to address what is the effect of the magnetic field-alignment along the jet. This makes an assumption that the jet is moving along the direction of the ion velocity during the interval identified as a jet. Authors should demonstrate how accurate this assumption is for few cases. They may consider the following:

a) Is there a goof de Hoffman teller frame for the jet structure? If there is not, why not, for example is the jet still accelerating when MMS crosses it? b) If there exists a good de Hoffman teller frame, how does the direction of the Hoffman teller frame velocity of the jet structure compare to the direction of the ion velocity? c) Are the any cases where the 4 spacecraft measurements can be used to determine the actual propagation direction of the jet as supposed to using the measured ion velocity?

These questions are relevant as for example in the simulations of Karimabadi et al., 2014 the direction of the jet motion appeared not to always align with the ion velocity but even sunward flows were seen adjacent to jet structures moving toward magnetopause. Did you observe any sunward flows in your statistics?

d) The pre-state of the magnetosheath field and plasma before the jet formation is likely to be very important for the subsequent jet propagation dynamics. It would be interesting to sort (for example using both IMF clock and cone angle) the jet events

based on the pre-IMF orientation before the radial turning. The current method of taking few minutes of the data before the jet may not be the truly "pre -state" of the magnetosheath depending on the shock geometry and how far the spacecraft is from the shock.

Minor comments:

Lines 10-11: Authors may consider citing recent paper by Nykyri et al., JGR, 2019 which discussed 14 spacecraft observations and the jet impact on substorm onset and showed that magnetosheath jets were associated with bursts of negative Bz in the magentosheath while IMF was northward. The DMSP spacecraft detected southward-like IMF erosion of the dayside magnetopause during jet observations during northward IMF, supporting evidence for jet-produced dayside reconnection.

---

## Author Comment (AC1) · 28 Nov 2019

**Response to the reviewers' comments**

First of all, we would like to thank the reviewers for their useful and constructive comments. They have helped us to substantially improve our manuscript. Below, the reviewers' comments are given in bold face and our answers are given in normal blue type. Page and line numbers refer to the original manuscript.

**Reviewer #1**

**The manuscript presents statistical results of how the angle between the local magnetic field and velocity vectors varies during magnetosheath jet events. This work has been motivated by recent simulations and case study observations. Given the myriad of impacts on Earth's magnetosphere that magnetosheath jets can have, understanding their propagation from their bow shock origin to the magnetopause is important and has largely been an open question in this particular topic within the solar wind - magnetosphere dynamical coupling. The methodology and results are clear and well presented and the results quantitatively align with previous theoretical interpretations of previous work, lending the statistical weight to these. I recommend publication subject to the authors addressing a number of minor issues.**

**General Comments:**

**[Reviewer 1 Comment 1] It was not clear to the reviewer whether the angles used (phi) were limited to be the acute angles (0-90 degrees) between the vectors. The figure limits throughout suggest this may be the case. However, while perhaps unlikely, it might be possible under certain configurations that a jet could bend the magnetic field lines back on themselves significantly resulting in angular deflections greater than 90 degrees which this analysis would not capture. This would result in the wrong angle being measured in the deflected regions. The authors should check that no greater than 90 degree deflections within the jet from pre/post occur in the dataset. If they do, the authors will need to re-do the analysis using the full angle between the vectors. They may wish to counteract the effects of the sign of Bx, which would lead to two separate populations in the data corresponding to either side of the heliospheric current sheet, by the average pre/post interval sign of Bx into account when calculating the angles for each event.**

The reviewer is right; the angles phi are limited to acute angles ($0° - 90°$). As suggested by the reviewer, we have split the dataset into two subsets based on phi being above or below 90° before and after the jet intervals. We have then performed the superposed epoch analysis on both subsets, without the restriction to acute angles. In both cases, however, the results are almost identical to the results shown in Figure 3. There are no indications of larger magnetic field deflections. We have added some explanations on this issue in line 6 of page 5.

**[Reviewer 1 Comment 2] In several cases, the authors quote median values as well as standard deviations. However, commonly a standard deviation is a difference from the mean value rather than the median. Medians are appropriate here as the mean is likely to be affected by outliers. So to would even a standard deviation about the median be affected. Quoting the lower and upper quartiles would be more appropriate throughout.**

We agree with the reviewer and have replaced all standard deviation values in the manuscript by upper and lower quartiles. For Figure 7 (corresponding text starting on page 9 line 9), we now use the inter-quartile range instead of the standard deviation. This does not change the results qualitatively, nor does it change the conclusions.

**[Reviewer 1 Comment 3] The abstract did not make it clear that there is a significant trend in deflection angles with jet speed. This is a key result of the paper and should be made more prominent in the abstract.**

We agree with the reviewer and have added a sentence to the abstract to make this result more prominent.

**Specific Comments:**

**[Reviewer 1 Comment 4] Page 1 Line 2 - The authors should also briefly comment on other scenarios such as that proposed by Karlsson et al. [2018, Ann. Geophys., https://doi.org/10.5194/angeo-36-655-2018] concerning SLAMS transmission through bow shock ripples which have recently been shown in Vlasiator simulations [Palmroth et al. [2018, Ann Geophys, https://doi.org/10.5194/angeo-36-1171-2018].**

We assume that this comment refers to page 2 line 2 instead of page 1. We agree with the reviewer and have added the scenario to the introduction, as suggested.

**[Reviewer 1 Comment 5] Page 4 Line 14 - Do the authors have an estimate on the number of independent jets observed, taking into account those that were observed by the same spacecraft?**

As stated in the manuscript, the MMS spacecraft are very close together at apogee, and only around apogee they are in the magnetosheath. Hence, jets are almost always observed by all 4 MMS spacecraft simultaneously. Correspondingly, the number of independent jets contributing to the superposed epoch analyses should be just slightly higher than 2477 jets. This number corresponds to MMS 4, as indicated on page 4 line 14. It is the largest number of jet observations per spacecraft.

**[Reviewer 1 Comment 6] Page 4 Line 23 - It is unclear why twice the solar wind density is used as a comparison measure when the bow shock typically compresses the solar wind density by a factor 4. If the authors mean the enhancement in density was greater than twice the solar wind density they should so state and make this clearer in how this result is depicted in Figure 2.**

We think that there might be a misunderstanding here: "Twice the solar wind density" is used as a threshold value to identify magnetosheath intervals (see line 6 on page 3). This threshold value is sufficiently large to exclude solar wind intervals, but also sufficiently below the nominal factor of 4 in order to avoid excluding large intervals of magnetosheath data from the analysis. In line 21 on page 4 we just wanted to note with the reference to twice the solar wind density that the interval shown corresponds to the magnetosheath. We have modified this sentence to state this more clearly.

**[Reviewer 1 Comment 7] Page 6 Line 17 - While the pre/post interval angle does decrease with cone angle (a result of the different draping patterns), it does appear that there is a very slight difference**

**in the depth of the median deflections with cone angle. The authors should estimate these depths (the effect size) and the significance of any differences with cone angle more thoroughly.**

We agree with the reviewer. We have checked the maximum change M in median angles (effect size) with cone angle (Figure 4) and velocity V0x (Figure 5). The result is as follows:

Median cone angles of 20.9°, 39.8°, and 61.3° correspond with M = 14.6°, 13.9°, and 10.5°. Hence, there is a small difference in alignment effect strength between jets observed during large IMF cone angle conditions in comparison to jets observed during low and medium IMF cone angle conditions.

Median velocities V0x of -130, -175, -221, -293 km/s correspond with M of 8.9°, 11.3°, 14.7°, and 18.8°, respectively. There is a clear linear trend between these two quantities: M(V0x) = 0.8669° − (0.0612° s / km) * V0x.

These values are now all stated in the manuscript. In addition, we also state M for median phi angles as shown in Figure 3.

**[Reviewer 1 Comment 8]** **Page 8 Line 1-2 - Can the authors comment more on expected draping angles at MMS's location for jet events or at least cite previous statistical studies into IMF draping near the magnetopause?**

The expected or typical angles phi between B and V at the location of the MMS spacecraft are experimentally determined from all the MMS magnetosheath data. Overall, the median angle is 59.2°, as stated in the manuscript. We think that this is by far the best and most precise way to obtain the expected value of phi for comparison. This issue is discussed in further detail in response to comment 2 by reviewer 2, below.

**[Reviewer 1 Comment 9]** **Page 8 Line 9 - The jet identification method does not necessarily mean a deflection towards the Sun-Earth line. Given the criteria, it could possibly have been the case that the y and z components of V similarly increased as Vx does which would not result in a deflection.**

Indeed, Vx, Vy, and Vz may increase simultaneously. Hence, jets identified with the stated criteria do not necessarily have to feature a flow deflection: we have modified the text in the manuscript accordingly. However, based on earlier results by Karlsson et al. (2012), Archer and Horbury (2013), and also Plaschke et al. (2013), we know that jets are typically deflected towards the Earth-Sun-line. Hence, we can assume that this applies also to the jets analyzed here.

**[Reviewer 1 Comment 10]** **Page 8 Line 20 - The authors should perform simple estimates of expectations given the picture in Figure 1 i.e. from a purely geometric point of view, ignoring any resistive forces, how much deflection would be expected for the set of observed jets and draping angles purely by the jet's flow locally advecting the field lines. To what extent could e.g. magnetic tension forces slow the jet's motion thereby reducing the deflection etc. This would bring into context the results and interpretation more clearly.**

The simplistic sketch shown in Figure 1 is good to visualize why there could/should be an alignment between B and V at all, but we doubt that we can draw any quantitative expectations from it. While the evolution of jets is outside the scope of this paper, we have found out in responding to comment 2 by reviewer 4 that almost all jets have a good deHoffmann-Teller frame. Hence, based on this

observation we may conclude that the deceleration of jets, e.g., due to magnetic tension forces, should not be significant.

**[Reviewer 1 Comment 11] Page 9 Line 9 - The authors should also mention another statistical method which might be adopted - computing individual deflection depths based on the (average of) pre/post jet intervals and estimating the distribution of deflection angles from this, rather than distributions of absolute angles. I am not advocating this be done for this manuscript, merely suggested as future work.**

We have tried out the method suggested by the reviewer. We took the averages of phi in the pre-jet intervals, and then computed the differences to the minimum phi values within the jet intervals, for every jet. The result is a very broad distribution peaking between 45° and 50°. However, this distribution does not convey any new information. It just means that the random changes in phi within jets (and also outside of jets) will be usually more significant than the systematic changes revealed by the superposed epoch analyses. In agreement with this interpretation, the distribution of differences to the minimum phi values within the pre-jet intervals is almost as broad and maximizes at almost the same angles: between 40° and 45°.

**Technical Corrections:**

**[Reviewer 1 Comment 12] Page 4 Line 12 - The statement "We require Vx to be negative within jet intervals and surpass half of its value at t0 within both pre- and post-jet intervals" is a little confusing and I would suggest the authors instead of discussing a negative number surpassing a threshold in the pre/post interval, instead talk about the absolute value dropping below said threshold.**

We agree with the reviewer and have changed the sentence accordingly.

**[Reviewer 1 Comment 13] Page 5 Line 4-5 - Please make it clear that values at each individual time are used, showing the same symbols as in the figures i.e. B(t) V(t). This will help contrast later with the other angles used.**

We agree with the reviewer and have made the appropriate changes.

**[Reviewer 1 Comment 14] Page 6 Line 6 - Please make clear that the magnetic field here is still taken at each individual time B(t).**

We agree with the reviewer and have made the appropriate changes.

**Reviewer #2**

**This paper presents the effects of magnetosheath jets on magnetosheath magnetic fields based on epoch analysis. The results clearly show that the magnetic fields tend to be aligned with the velocity of magnetosheath jets. The authors also discuss the consistency and inconsistency with the previous**

**case studies and simulations. That discussion looks good to me. Thus I only have some minor comments as shown below.**

**[Reviewer 2 Comment 1] Line 6 of Page 6: You use the angle between B(t) and V0 to do epoch analysis but can you explain why you did this? The angle between B(t) and V(t) is easy to understand while B(t) and V0 usually occur at different time/location, so it is difficult for me understand why you compare these two vectors. In addition, in figure 3, the alignment effect is more significant shown in black lines than red lines. Do you have a good explanation of that?**

The velocity V0 corresponds to the maximum dynamic pressure measured within a jet. We take this velocity vector as an indication of the overall jet propagation direction. From a previous study (Plaschke and Hietala, AnGeo, 2018) we know that the ion velocity vector direction changes slightly inside jets due to vortical plasma motion. As a result, there is a range of velocity vector directions from which we could choose or compute the most accurate to represent the overall direction of propagation of a jet. We think that V0 serves that purpose best, and is best suited as measured reference direction to compare magnetic field directions against. We have added an appropriate note to line 6 on page 6.

Regarding the black and red lines in Figure 3: Around tn=0, both curves are obviously the same. Before and after, V(t) should deviate more from the Sun-Earth-line than V0, as jets are usually propagating more in anti-sunward direction in comparison to the surrounding plasma. Consequently, the angles between B (draped IMF, almost tangential to the magnetopause) and V at tn=-2 and 2 may be a little smaller on average than between B and V0 at the same two normalized times. Hence, the differences in phi_(B,V) will also be smaller than in phi_(B,V0): The alignment effect will be seen more prominently in the black than in the red curve.

**[Reviewer 2 Comment 2] Figure 3 or Lines 1-3 of Page 7: I saw you discussed about why the angle between B and V is approx. 60-70 degree at t(-2) or t(2). You said that this value indicates "the typical angles between magnetic field and plasma flow directions in the subsolar magnetosheath". Is there any reference showing that typical angle? In addition, do you think the locations of MMS probes (e.g., closer to magnetopause vs. closer to bow shock) affect that angle in the background magnetosheath plasma? Furthermore, magnetosheath jets may evolve in the magnetosheath as they propagate from bow shock to magnetopause, do the locations of MMS probes also affect the angle change with magnetosheath jets? Is it possible to briefly discuss about that with your current database?**

We discuss this issue partly above, in response to comment 8 by reviewer 1. To our knowledge, there is no good reference stating this angle, but we compute a typical/expected angle phi ourselves (59.2°), based on all MMS magnetosheath observations.

We do indeed think that the typical angles phi should change as a function of the location between magnetopause and bow shock, and also between subsolar and more flank locations, as a function of the combined draping and flow patterns. This is now stated on page 8 line 2.

As jets evolve between bow shock and magnetopause, also the angle distributions/changes should evolve. Indeed, we find the alignment effect to be dependent on the relative distance of the observing spacecraft between the magnetopause and the bow shock (r_rel): The effect is stronger closer to the bow shock than in the central sheath or closer to the magnetopause, as we discuss in our answer to comment 1 by reviewer 4.

**[Reviewer 2 Comment 3]** Lines 16-17 of Page 8: ". . . much smaller than seen in simulations by Karimabadi et al., (2014)". You attributed it to the 2D not 3D simulation in the previous simulation. Their simulation seems to be done in the XY plane and do you agree that if you do an epoch analysis in that 2D plane, you will obtain the similar result as what their simulation obtained?

We do not agree: We did the 2D epoch analysis using only the x and y components of B and V, as suggested, and the results we obtain are similar to the results shown in the manuscript. The differences to the simulation results by Karimabadi et al. (2014) persist.

**[Reviewer 2 Comment 4]** Lines 4-9 of Page 10: The second conclusion says the statistical results got smaller angle change than the previous simulation got; the third conclusion says that the large fluctuations in sub-jets may mask the decrease in phi. If there is a way to remove the effects by sub-jets (you don't have to do that), do you think the decrease in phi will be comparable to what the previous simulation shows? Or you still consider 2D vs. 3D is a important issue here?

We do not think that the systematic decrease in phi will become comparable to simulation results once the random fluctuations are removed: The superposed epoch analysis does a good job in removing the fluctuations and, still, the effect strength (decrease in phi) is far smaller than what is seen in the simulation, where the magnetic and velocity fields become essentially fully aligned. The 2D vs 3D point remains an important issue here, in our opinion, also because recent simulation activities of jets in 3D unravel much more complex magnetic field structures.

**Reviewer #3**

**GENERAL COMMENTS:**

This paper is well written, and presents the results of a large statistical study of the alignment of the flow direction associated with a magnetosheath high speed jet and the local magnetic field in a clear and concise fashion. The authors show that while there is a deviation of the local magnetic field direction such that it becomes more aligned with plasma flow direction of the jet, this is not a large effect on a statistical level, contrary to recent modeling results. While obtaining a deeper understanding of the general nature of the structure of the jets is important, there is not clear connection in the text as to why this particular aspect of the jet is important for further understanding local processes in the magnetosheath or how the alignment of the magnetic field and velocity vector impacts interactions of the jet with local ambient plasma and the magnetopause.

**SPECIFIC COMMENTS:**

**[Reviewer 3 Comment 1]** Page 3: The "30°-wide cone centered at Earth and open to the Sun" used for selecting jet intervals simply corresponds to MLTs of 11-13 hours, correct?

It corresponds to about 10 – 14 hours in LT. We have added this to the description of the selection criterion.

**[Reviewer 3 Comment 2]** Page 6, Line 15: Along with intrinsic conditions or upstream solar wind possibly contributing to the limited alignment effect, does spacecraft trajectory through the jet structure have any effect on the observations of the jet?

The exact sequence of angles phi as a function of normalized time will be different depending on where a jet is actually crossed by the spacecraft. The question is, however, if/how these differences are reflected in the superposed epoch analysis results. Unfortunately, it is impossible for us to assess this effect, because we cannot determine where the MMS spacecraft cross the jets, due to the small spacecraft separations.

**[Reviewer 3 Comment 3]** Page 6, Lines 16 – 18: Looking at Figure 4, there does appear to be a slight dependence on IMF cone angle. When you look at the percent change relative to the phi_B,V0 level at t = -2 for the different cone angle bins (i.e., looking at the change in the angle at t = 0 after subtracting out as an offset the value of the angle at t = -2 for each cone angle bin), is any dependence of cone angle seen?

Yes, there is a slight dependence of the B/V alignment effect on IMF cone angle. We quantify this effect now in the manuscript. For more details, see our response to comment 7 by reviewer 1.

**[Reviewer 3 Comment 4]** Page 6, Lines 10-14 and Page 9, Line 6: Can you show another jet example that has the more common feature of a smaller change in the alignment of B and V?

We think that there might be a misunderstanding here. The jet shown as an example is just special in that phi is low at exactly tn=0. However, the variability in phi seen in that example over the entire jet interval is very common. Many other jets also feature low phi values, but maybe not at tn=0. In that sense, the example shown is not uncommon at all.

**[Reviewer 3 Comment 5]** Since solar wind conditions were used for this statistical study, are there any indications that other upstream conditions may be related to the largest changes in alignment of B and V?

We have tested this and conclude that there are no indications that the upstream conditions (velocity, density, dynamic pressure, magnetic field, Mach numbers) at jet observation times are related to large systematic changes in alignment. We state this now in the results and conclusion sections.

**[Reviewer 3 Comment 6]** In the discussion section, more results are presented on the change in plasma velocity and the standard deviation of the angle between B and V, which is helpful in interpreting the superposed epoch analyses of the changes in alignment between B and V in the core of the jets. However, there isn't much discussion on the implications and consequences of the limited alignment effect seen for the majority of the jets. What does the small change in alignment mean for interactions with the local plasma or the magnetopause? Does it particularly matter, and if so, why? More discussion on this would be beneficial for grounding the results in the broader context of the studies mentioned in the introduction section.

The magnetosheath plasma and fields are the input to any interaction with the geomagnetic field at the magnetopause. An important question in this respect is: How do jets affect the magnetosheath plasma and fields? This question is partially answered in this paper by investigating to which degree

magnetic and velocity fields become aligned due to the passage of jets. Jet-induced changes in the magnetic field are also expected to have repercussions with respect to magnetosheath current sheets and reconnection within the magnetosheath and at the magnetopause. We have added this discussion at the end of the introduction section.

While jets occur very frequently, many of them are small and may not have, by themselves, significant effects, e.g., on B/V alignment. We say this on page 9, in the paragraph starting on line 4: The superposed epoch analyses yield an average picture based on several thousand jets of different sizes and characteristics. Further studies are required to show which jets are effective in changing their environment.

**TECHNICAL CORRECTIONS:**

[Reviewer 3 Comment 7] Page 2, line 8: change "on ground" to "on the ground" Page 2, line 9: add "of the magnetopause" after "surface" Page 8, line 1: change "extend" to "extent" Page 8, line 8: change "in anti-sunward" to "in the anti-sunward" Page 8, line 12: change "similar" to "similarly"

Thank you for noticing these mistakes. They have been corrected in the revised manuscript.

**Reviewer #4**

**Overview:**

**Paper focuses on studying if there is statistical, observational, evidence supporting the simulations of Karimabadi et al., 2014 that magnetosheath jets make the ambient magnetic field more aligned with the jet velocity. Study uses data (FPI for ion moments and FGM for the magnetic field) from the four Magnetosphere Multi-Scale (MMS) spacecraft between September 2015 and May 2017. The main conclusions by the authors, obtained based on superposed epoch analysis of the pre and post jet angles between ion velocity and magnetic field, is that while jets generally modify the magnetic field, the alignment of the magnetic field with the jet flow is relatively small. They suspect the discrepancy may be due to the 2-D simulation geometry, while the real nature is 3-D.**

**Paper addresses a compelling topic as understanding of the jet formation, structure and propagation thorough magnetosheath and their subsequent impact on the magnetopause will help address magnetospheric response to the dynamic driving by the solar wind and may be even relevant for the long-standing "internal" vs "external" substorm triggering debate. The authors have compiled an extensive data base for addressing this topic using MMS data and I think the paper is suitable for publication after some revisions. I also think that the manuscript would strongly benefit from some additional analysis before clear conclusions can be made. I would recommend the authors to perform some additional analysis (which should not take too long) and address the following in the revised manuscript:**

**Main comments:**

**1. The superposed epoch analysis uses normalized time and organizes the data on pre- and post jet intervals based on this time. However, the number of single spacecraft measurements do not provide information on the 3-D jet structure, magnetosheath structure or how the spacecraft might have crossed the jet. Because the spacecraft are nearly stationary when compared to jet propagation**

speed, the "pre jet" and "post jet" time intervals may correspond to the vastly different spatial regions with respect to jets due to different 3-D field geometries that can arise due various factors, e.g., how spacecraft crosses the jet, the distance to the jet formation region (is the jet accelerating, moving at constant speed or decelerating), how does the jet dynamic pressure relate to the ambient magnetic field pressure in the magnetosheath, to the distance to the magnetopause and due to different spacecraft z-component in GSM coordinates. Note that during 2015-2017, close to the spring and fall equinox times, the MMS GSM z-coordinate in the dayside magnetosheath can be substantial so this list will likely include several high-latitude magnetosheath observations close the dayside high-latitude magnetopause.

I would recommend the authors perform and address the following in the revised manuscript:

[Reviewer 4 Comment 1] a) Sort all of the identified jet intervals based on the distance to the model magnetopause and model bow shock calculated using prevailing solar wind condition during the jet observations.

b) Show a distribution of the MMS z-position during jet observations and study the dependence of the MMS z-coordinate and distance to the magnetopause and bow shock on the deviation of the "pre jet" and "post jet" angles. Study how the ratio of the local dynamic jet pressure and the pre-jet magnetic field pressure varies as function of distance between magnetopause and bow shock and how this affects the angles.

We have calculated the relative locations r_rel of jet observations between the magnetopause (r_rel = 0) and the bow shock (r_rel = 1) using the Shue et al. (1998) and Merka et al. (2005) models. Most of the jets pertain to a central sheath location. There are however also jets observed very close to the magnetopause (1856 jets at r_rel < 0.25) and very close to the bow shock (797 jets at r_rel > 0.75). We state this at the end of the data and methods section, where we now introduce the r_rel parameter.

There is indeed a dependency of the alignment effect on r_rel. The effect is notably stronger closer to the bow shock than in the central sheath or closer to the magnetopause. Interestingly, although MMS observations closer to the bow shock are associated with higher solar wind dynamic pressure values (as expected), the alignment effect itself is not (strongly) dependent on that parameter. We have added two figures to the paper to show this and expanded the text at the end of the results section accordingly. We have also added some discussion on this dependency at line 3 of page 9 (discussion section) and have updated our conclusions section accordingly.

We have also taken into account the MMS z-positions during the jet observations. However, there is no discernable trend in the deviations of the angles phi with respect to the z-coordinate. There is also no discernable trend with respect of the distance R of the spacecraft from the Earth-Sun-line. We think that our initial restriction to the subsolar magnetosheath on creating the jet data set does not allow us to study changes in phi angle behavior with respect to z or R in a meaningful way.

[Reviewer 4 Comment 2] c) The analysis uses the velocity vector as measured by the MMS to calculate the angles for pre-jet, jet and post jet intervals to address what is the effect of the magnetic field-alignment along the jet. This makes an assumption that the jet is moving along the direction of the ion velocity during the interval identified as a jet. Authors should demonstrate how accurate this assumption is for few cases.

They may consider the following:

**a) Is there a good de Hoffman teller frame for the jet structure? If there is not, why not, for example is the jet still accelerating when MMS crosses it?**

**b) If there exists a good de Hoffman teller frame, how does the direction of the Hoffman teller frame velocity of the jet structure compare to the direction of the ion velocity?**

We have computed the deHoffmann-Teller frame for all jets, using the data between normalized times -1 and 1 (jet intervals). Indeed, the analysis shows that there is a good dHT frame for basically all the jets, indicating that they are coherent structures with quasi-stationary magnetic field and velocity patterns. Consequently, jet acceleration or deceleration should not be strong. The direction of the frames $V\_dHT$ is also mostly close to V0. In 44% of the jets, the angle between $V\_dHT$ and V0 is below 10°, and in 85% of the cases, it is below 20°. We now include a note in the manuscript (page 6, line 6), indicating this. The results with respect to the changes in phi do not differ much, whether we take $V\_dHT$ or V0.

**[Reviewer 4 Comment 3] c) Are the any cases where the 4 spacecraft measurements can be used to determine the actual propagation direction of the jet as supposed to using the measured ion velocity?**

As the MMS spacecraft are close together, any timing analysis yields only the propagation velocity of local internal structures/current sheets projected onto the normal vectors of those structures. And these normal vector directions will vary a lot within jets: In Plaschke et al. (2017), the structure velocity is denoted with Vs. In Figure 5 of that publication Vs is shown in panel (b) and the ion velocity is shown in panel (a). As can be seen, the ion velocity does not change in direction much; it mostly points in –x-direction. Vs, instead, is wildly fluctuating due to the rich internal structure of the jet considered. This is in agreement with the strong variations of phi within individual jets reported in the manuscript under review.

From this observation we can conclude that individual samples of Vs will not yield the propagation direction of the jets. However, it is not unimaginable that a sufficiently big sample of different Vs pertaining to one jet may allow sometimes for a determination of the jet propagation direction via deprojection.

**[Reviewer 4 Comment 4] These questions are relevant as for example in the simulations of Karimabadi et al., 2014 the direction of the jet motion appeared not to always align with the ion velocity but even sunward flows were seen adjacent to jet structures moving toward magnetopause. Did you observe any sunward flows in your statistics?**

Karimabadi et al. (2014) observe sunward flows in the vicinity of jets, as they pass by. With the MMS data set, this question cannot be addressed, because the MMS configuration is too small to provide context observations outside of jets and simultaneous observations of the jets themselves. However, with THEMIS multi-spacecraft observations of jets, this question can be answered and, indeed, already has been answered: Plaschke et al. (2018) investigate flow patterns in and around jets, based on THEMIS multi-spacecraft measurements. In that paper/study, no sunward flows were observed, neither in the superposed epoch analysis results nor in individual jet observations.

**[Reviewer 4 Comment 5] d) The pre-state of the magnetosheath field and plasma before the jet formation is likely to be very important for the subsequent jet propagation dynamics. It would be**

interesting to sort (for example using both IMF clock and cone angle) the jet events based on the pre-IMF orientation before the radial turning. The current method of taking few minutes of the data before the jet may not be the truly "pre -state" of the magnetosheath depending on the shock geometry and how far the spacecraft is from the shock.

There is no systematic change in B/V alignment with pre-state instead of tn=0 shear angles, but there is some dependency on the cone angle "pre-state". If we evaluate the IMF cone angles 20 min before tn=0 instead of at tn=0, we obtain noticeably larger maximum angular changes M for low cone angle events: M= 17.3° instead of M=14.6°. As suggested by the reviewer, this could be due to a different pre-state of the magnetosheath when the jet is generated. We state this now in line 19 of page 6 and in line 3 of page 9.

**Minor comments:**

**[Reviewer 4 Comment 6] Lines 10-11: Authors may consider citing recent paper by Nykyri et al., JGR, 2019 which discussed 14 spacecraft observations and the jet impact on substorm onset and showed that magnetosheath jets were associated with bursts of negative Bz in the magentosheath while IMF was northward. The DMSP spacecraft detected southwardlike IMF erosion of the dayside magnetopause during jet observations during northward IMF, supporting evidence for jet-produced dayside reconnection.**

We agree with the reviewer and cite the paper in a new paragraph inserted at the end of the introduction section.